# Challenges and Current Research Trends for Vernacular Architecture in a Global World: A Literature Review

José María Fuentes Pardo 

BIPREE Research Group, Universidad Politécnica de Madrid, 28040 Madrid, Spain; jm.fuentes@upm.es;
Tel.: +34-910670936

**Abstract:** Vernacular architecture can be defined as a type of regional construction influenced by geography, available materials, climate, traditions, and culture that is produced by non-experts through knowledge transmitted and enriched from one generation to the next. In addition to their heritage value, traditional buildings are usually considered a model of sustainability in the use of available resources, with a minimal negative environmental impact, minimization of costs, and a reduction of energy demand. In the current context of the globalization of construction techniques and high requirements of comfortable dwellings, the preservation of vernacular architecture means a major challenge, given that this kind of architecture represents more than 75% of the world's existing buildings. Based on a study of selected peer-reviewed literature indexed in the Web of Science for the period between 2000 and 2022, this paper qualitatively analyzes the current areas of research on vernacular architecture, with particular attention to the scope of the studies, traditional building materials and construction techniques, preservation problems and solutions, climate adaptation, and the reuse of abandoned vernacular buildings. In order to achieve the goal of preserving vernacular architecture in the coming centuries, research should continue in interdisciplinary teams by promoting fieldwork in under-studied regions and incorporating modern materials and solutions in old vernacular buildings to satisfy current comfort standards without excessively changing their essential features.

**Keywords:** vernacular architecture; traditional buildings; research; trends



## 1. Introduction

The interest in vernacular-built heritage emerged in 19th-century England as a critical response to the process of industrialization [1]. Supported by John Ruskin's theories, the Arts and Crafts movement found in the vernacular a way to return to craftsmanship through respect for the nature of materials and the rescue of traditional methods. With the development of anthropology and ethnography as academic disciplines, traditional constructions also became a valuable resource for the understanding of human communities and their cultural practices.

It was Rudofsky who first used the term 'vernacular' in an architectural context as a synonym for 'anonymous' or 'spontaneous', following the exhibition 'Architecture without Architects' held at the New York Modern Art Museum in 1964 [2]. In the same period, Rapoport studied the form of dwellings for different vernacular communities to identify its driving factors, arguing that social and cultural factors have more influence on the creation of the form than physical and environmental ones [3].

The vernacular evokes those traditional buildings typical of a certain region and is profoundly influenced by geography, available materials, climate, traditions, and culture that are produced by non-expert "ordinary people" through knowledge transmitted and enriched over time from one generation to the next [4]. However, identifying vernacular architecture with the most typical constructions of a given region is only a part of the picture. The term vernacular can be extended to include common residential dwellings in

towns or cities, preindustrial buildings, commercial facilities, and collective buildings or infrastructures, among others [5].

The main attributes that characterize vernacular architecture are: (i) the use of local materials; (ii) a planimetric design adapted to the climatic and topographical conditions; (iii) the use of construction techniques and aesthetic resources, transmitted and readapted over time; and (iv) the active participation of users and local craftsmen in its design and construction (Figure 1). The result is a highly varied architecture (even within the same locality or region) as a consequence of its adaptation to the social conditions of dwellers, the planned use, the technology, and the social context in which it was conceived [6,7].

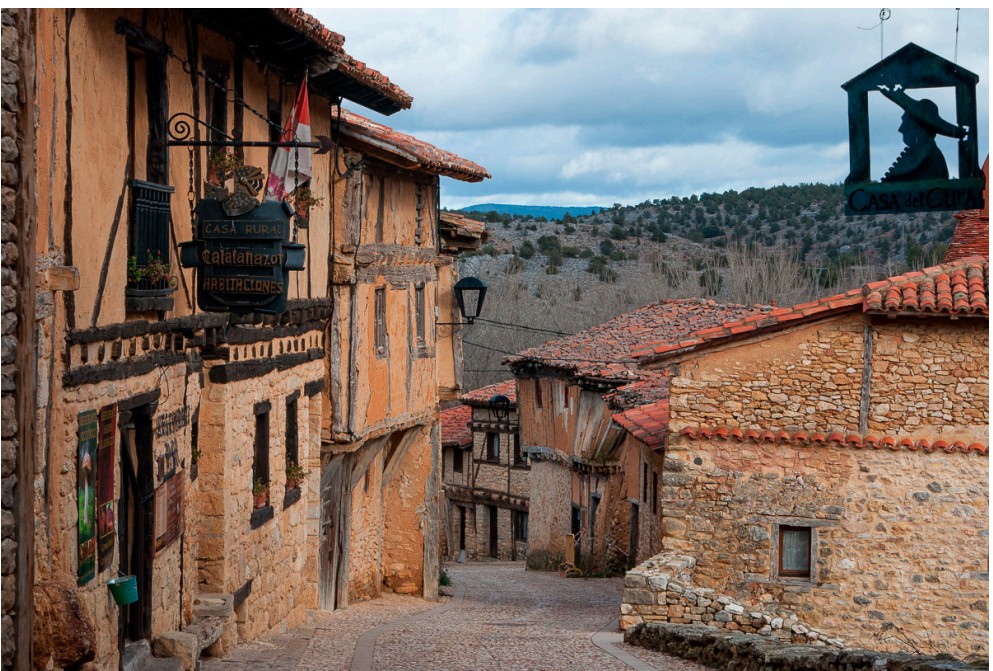

**Figure 1.** Vernacular houses in Calatañazor, Central Spain.

Vernacular architecture undoubtedly possesses aesthetic, ethnological, and anthropological values. The second half of the 20th century was characterized by the emergence and subsequent consolidation in the international context of several organizations concerned with the protection of built heritage (i.e., the National Trust, English Heritage, the Society for the Protection of Ancient Buildings; the Getty Conservation Institute; the Instituto Centrale per il Restauro (ICR) or the International Council on Monuments and Sites (ICOMOS)). The efforts of these organizations and other public institutions, such as UNESCO or the Council of Europe, led to the enactment of different standards or "charters" for the conservation of ancient buildings. The Venice Charter for the Conservation and Restoration of Monuments and Sites of 1964 incorporated modest architectural constructions of outstanding documentary value into the category of monuments, with an emphasis on rural-built heritage [8]. The Charter on the Built Vernacular Heritage ratified by ICOMOS in 1999 [9] extended the objectives of the Venice Charter and provided fundamental principles for the protection and enhancement of vernacular architecture. Thus, the built heritage of a region is not only evidenced in prominent monuments or large public works but also in common dwellings and other functional or productive buildings.

Vernacular architecture is a distinctive feature of each region of the world, as it is consistently adapted to respond to climate, topography, local resources, and social lifestyle [10–12]. Due to globalization in the construction techniques and the living patterns experienced in recent decades, local architectural distinctiveness has vanished, and developing countries welcomed the importation of modern industrialized construction systems, which do not always adequately respond to the needs of users [13]. Thus, despite the fact that vernacular

architecture comprises more than 75% of the world's existing buildings, it is poorly studied, and its preservation raises big concerns and represents a major challenge.

Although the vernacular architecture of vast areas of the planet remains unstudied, certain regions are currently living a resurgence of interest in vernacular architecture and in the use of locally produced renewable materials in contemporary construction. Whereas in the past, raw materials were used to provide shelter and meet the cultural, social, and economic needs of individuals and communities, today, they can be used as a strategy to support social engagement, sustainable development, and cultural continuity [14]. At the same time, the use of vernacular strategies in modern architectural design can also serve to fulfill basic ecological principles of energy efficiency and move towards resource-friendly solutions [15].

Based on the result of many years of experience, vernacular buildings are generally less energy-demanding than modern construction systems, and traditional construction techniques are usually good examples of climate resilience and adaptation (i.e., thick walls to provide thermal inertia, shading and natural ventilation solutions, resistant roofing materials and wide eaves in snowy or rainy areas, etc.). Resilient design strategies are particularly important in extremely hot climates to prevent overheating, and adequate natural ventilation of the buildings avoids the use of air-conditioning systems to achieve acceptable thermal comfort [16,17].

Climate change effects, such as extreme temperatures, high rainfall, severe floods, or landslides, multiply the risks of climate-induced disasters on the vernacular heritage in certain regions. In addition, climate change means other risks for rural areas, such as crop failures, loss of cultivable lands, depopulation, maladaptation, and loss of knowledge and intangible values, resulting in the loss or abandonment of vernacular buildings and traditional construction techniques [18,19]. In this context, research on how vernacular buildings have progressively adapted to climate over time and on those characteristics that make them resilient to climate change must be a priority for today and for the near future.

The analysis of vernacular buildings before they deteriorate or fall into ruin is a useful documentary resource for contemporary architecture [20–22]. Traditional architecture is usually considered a model of sustainability in the use of the available resources, with minimum negative environmental impacts, the reuse of materials, and a reduction of energy demand [23–25]. On the other hand, the role of this architecture could be connected with new forms of ecological production, adapting and reusing certain constructions and infrastructures which, in many cases, have been obsolete for decades, in order to implement certain agricultural or livestock practices in a sustainable way [26,27].

However, the sustainability of vernacular architecture faces important challenges at present. On the one hand, the extraction of materials (stone, timber, etc.) sometimes generates significant environmental impacts, and the energy sources used in their production or to heat the buildings (i.e., burning peat, coal, oil, or wood) are not always renewable. On the other hand, the design and functionality of vernacular dwellings built several decades or centuries ago are not always adapted to the current family structure and/or to the comfort requirements of today's society. This dichotomy represents a major challenge for the conservation of vernacular heritage that must be rigorously investigated.

One of the essential characteristics of vernacular architecture is the active participation of anonymous people in the design and construction of the buildings. However, as vernacular architecture studies have emerged as an academic discipline and architects, conservation practitioners, legislators, or urban planners are increasingly aware and conscious of the value of the vernacular, there is a risk that, under the control of experts, vernacular buildings will lose their intrinsic popular character, away from local agents and a sort of natural evolution [28].

Vernacular architecture is often presented in the literature as an example of environmentally friendly design with undoubted microclimatic advantages. However, climatic issues are often of secondary importance in the design to factors such as behavioral or cultural influences [29]. In his book, Architecture for the Poor, Hassan Fathy proposes

that architectural practice should focus on improving the quality of life of ordinary people, based on a detailed study of living habits and intelligent use of vernacular materials and construction systems [30]. Similarly, C. Alexander, in his book The Timeless Way of Building, advocates architecture centered on individual or cultural human needs and in intimate fusion with nature as the basis of sustainability [31]. According to Alexander's ideas, vernacular architecture constitutes an intuitive, spontaneous, and natural design model whose development and evolution cannot be planned.

The sustainability of vernacular architecture is only possible based on a comprehensive understanding of the living habits of dwellers and the cultural roots that have motivated the particular design of the buildings [32]. In Islamic society, for example, privacy is as decisive for the design of vernacular buildings as climate adaptation or aesthetics [33–35]. Only by understanding the context surrounding vernacular architecture (i.e., community priorities, cultural aspects, etc.) will it be possible to preserve its essence and integrate the most significant features into modern architecture.

Based on a literature analysis in an international context, this paper addresses some of the current challenges that vernacular architecture is facing nowadays and tries to identify key topics and research areas that may need further study in the forthcoming years.

## 2. Methods

A systematic literature review was carried out to examine the main research trends concerning vernacular architecture for the last few years, following the method proposed by [36]. A document search was performed in the Web of Science (WoS) database using the terms ("Vernac*" OR "tradit*") AND ("archit*" OR "build*" OR "herit*") in the title, keywords, and abstract fields. The WoS database was selected as one of the most widely used publisher-independent global search engines for peer-reviewed literature. The search was conducted in July 2022 and focused on documents published between January 2000 and July 2022. A list of 2470 documents was retrieved from this initial search.

A selective process was subsequently carried out on the preliminarily obtained records using the inclusion/exclusion criteria shown in Table 1. Duplicate records, publications unrelated to the subject of this study, publications in languages other than English, and those with a full-text and not easily accessible were discarded.

**Table 1.** Inclusion and exclusion criteria for the literature selection.

| Inclusion | Exclusion |
| --- | --- |
| Date of publication between January 2000 and July 2022 | Date of publication out of range |
| Full-text available | Full text not available |
| English | Non-English |
| Topics: Materials and techniques; climate adap-tation; protection and conservation; opportuni-ties and risks; reuse of abandoned buildings | Does not directly correspond to any of the mentioned topics |

The search was restricted to publications in peer-reviewed journals (excluding books and conference abstracts) as representative examples of the highest-level scientific research. Non-English sources, although potentially providing interesting insights, were finally discarded to avoid misinterpretations that could arise from their translation.

After an individual review of the title and abstract of each of the documents obtained in the preliminary selection, the topics covered by them were determined. In accordance with the objectives of this work, the search was limited to those publications addressing one of the following topics: materials or construction techniques, climate adaptation, protection and conservation, opportunities and risks, and reuse of abandoned vernacular buildings. The search filters provided by the WoS database were used for this purpose. After applying

the inclusion and exclusion criteria, a list of 270 documents was finally selected for detailed review and analysis.

A conceptual framework with the main research topics on vernacular architecture by areas identified in this study is included in Figure 2.

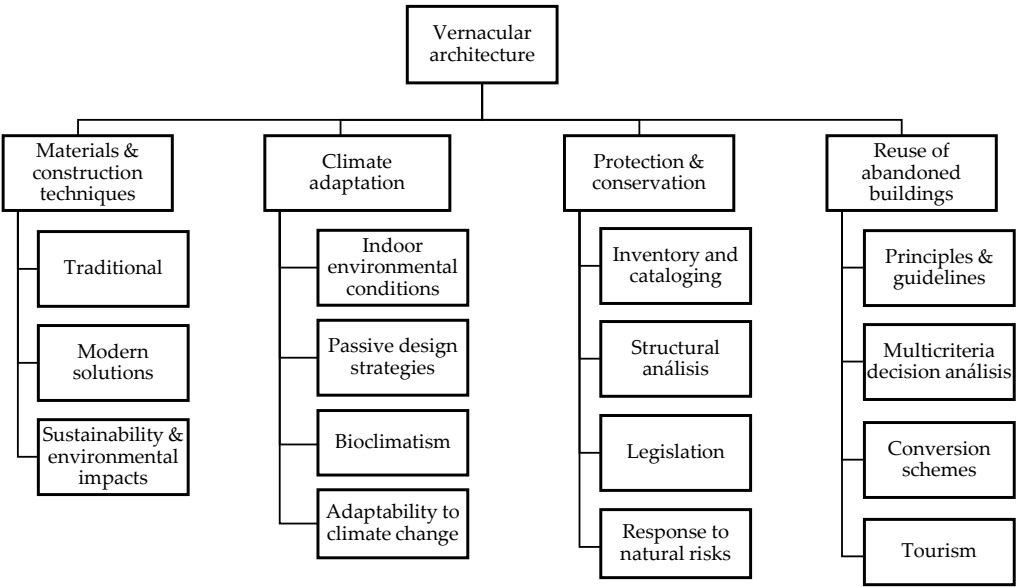

**Figure 2.** Research topics in vernacular architecture identified in the literature review.

The intention of this paper was not to attempt to compile all relevant research on vernacular architecture since this is an unaffordable task. That said, a significant sample of the available peer-reviewed literature documenting the major challenges and current research directions in vernacular architecture in an international context has been captured.

## 3. Results and Discussion

### 3.1. Aims of the Published Studies and Research Methods Used

In the last two decades, the published studies on vernacular architecture have significantly increased in different parts of the world, demonstrating a great interest in this topic by the building research community [23]. This increased trend is clearly observed in Asia and Europe and is likely to continue in the coming years (Figure 3).

However, it should be noted that the studies on vernacular architecture have been insufficient in many countries at the international level. From the total number of publications retrieved from the Web of Science search, only 8% correspond to regions located in Africa, South America, or Oceania.

Indigenous dwellings in Africa or South America are as varied as the numerous influences that have inspired them. The many different ethnicities make it difficult to speak of a single type of traditional housing, as each tribal village has its own morphology, iconography, and construction methodology, influenced and shaped by its social structure, religion, ethnic values, and local customs. Due to the consequences of colonialism, many of these architectural practices lost their relevance, giving rise to the predominance of Western styles and methods in contemporary construction. Unfortunately, the vernacular architecture of these areas has become progressively less attractive for homeowners, who often associate it with poverty. Consequently, the abandonment of indigenous architecture has resulted in a shortage of craftsmen skilled in the art of traditional construction, a reality that further dampens hopes for a renaissance [37].

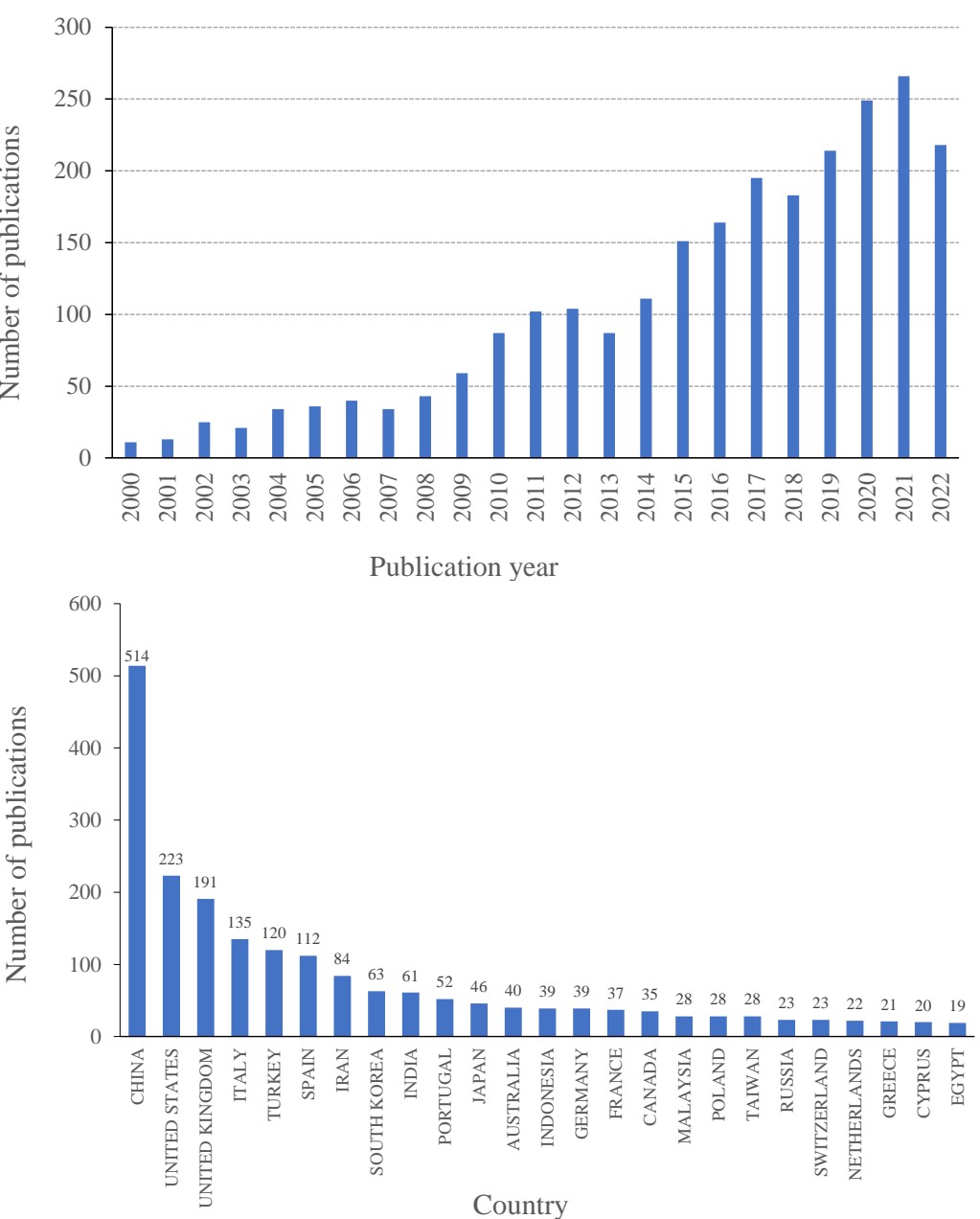

**Figure 3.** Publications on vernacular architecture indexed in the Web of Science for the investigated period. Above: distribution over time; Below: top 25 countries with the most published documents (data collected from the Web of Science in July 2022).

Vernacular architecture studies cover from vernacular architecture in arid or desert areas [38–41] to constructions typical of tropical or humid climates [42–44].

Vernacular houses, including specialized spaces such as open-air courtyards or cave dwellings, are the top research objects, as they are the most popular and easily accessible items that can be found anywhere in the world. The objectives of the research range from typological analysis and the study of architectural styles and forms to the analysis of the spatial patterns and influencing factors or the investigation of thermal passive or bioclimatic principles applicable to modern buildings [45–48]. Other types of vernacular constructions, such as traditional farm buildings; food manufacturing premises (wine cellars, flour mills, slaughterhouses, etc.); watermills, forges, or artisanal workshops; or infrastructures (ice houses; drystone walls, terraces, roads, or canalizations, among others)

have also been studied in some parts of the world, although they still should obtain stronger attention [49–54].

Research methodologies encompass quantitative and qualitative methods and combine literature and documentary sources with fieldwork.

Technologies based on Geographic Information Systems (GIS) can be exploited for the collection of information, geospatial analysis, and the integrated planning and management of existing buildings in a given region [45–58]. Terrestrial or unmanned aerial photogrammetric techniques, laser-scanning, and close-range photogrammetry, combined with BIM, reveal useful tools for surveying and documenting isolated buildings having an important historical value [59–61]. Some other modern techniques, such as deep learning, have also been successfully introduced to detect built cultural heritage from satellite images [62].

### 3.2. Traditional Building Materials and Construction Techniques

Traditional building materials, such as earth, stone, wood, or straw, are experiencing something of a renaissance against modern industrially produced materials, such as reinforced concrete, ceramic bricks, plastic, glass, or steel, in some parts of the world, since the manufacturing process of the latter is energy-intensive, generates a high carbon footprint and other pollutants, and involves high economic and transport costs [63,64]. Traditional building materials also offer other undeniable advantages, such as local availability, resistance to weather conditions, and better reintegration to the environment when buildings are demolished, albeit, on the other hand, building with traditional materials is usually a more complex, time-consuming, costly, and labor-intensive process.

In view of the above, environmental concerns have generated interest in research into new natural building materials, such as hempcrete [65,66], cob [67], earthbags [68], straw bale construction [69], compressed earth blocks [70], or rammed earth [71,72], among some others. However, the use of an environmentally friendly material in a certain local area may be a threat in another one. For example, bamboo, with its high natural tensile strength and rapid growth rate, may seem like a solution to replace wood. However, excessive demand can lead to the destruction of forests to make space for bamboo plantations, as is currently happening in certain areas of China [73]. The massive use of wood or stone can also have adverse effects on the environment in terms of deforestation or the intensive exploitation and subsequent abandonment of stone quarries [74]. To avoid such effects, the exploitation of natural resources to produce construction materials must be properly planned and regulated.

The existing literature on the physical and mechanical characterization of vernacular materials such as raw earth or dry-stone masonry is quite vast [75–79]. The use of numerical models and experimental testing with scale prototypes can be a helpful tool to evaluate and simulate the structural behavior of certain building components (i.e., masonry walls, timber structures, bridges, etc.) [80–82].

The addition of natural fibers or agricultural wastes to earthen materials is a common practice that can be found in both traditional and contemporary building systems [83–85]. A recent study by Paul et al. [86] recapitulated the impact of the incorporation of a great number of natural materials on the physical, mechanical, hydrothermal, and durability properties of earth-based composites. This review demonstrates that soil reinforced with natural fibers usually shows better strength properties compared to synthetic fibers because of their higher surface roughness, but cellulosic fibers can cause durability concerns due to their hydrophilic nature.

The performance of traditional plaster and render materials is also a recurrent research topic. Abundant literature on traditional gypsum or lime-based plasters [87–89] or surface-treated or fiber-reinforced earth coatings [90–92] has been published in recent years. Local coating materials, such as 'sarooj', a cementitious material used in South Asia and the Arabian Peninsula and obtained by the traditional calcination of raw clayey earth, have also been studied in order to be produced with modern technology and under controlled conditions [93].

The characterization, environmental performance, embodied energy, and cost of eco-efficient traditional and innovative insulation materials, such as cotton, cellulose, hemp, mineral or vegetable wool, and recycled rubber or cork, is also an area of research currently in vogue [94–96]. Modern solutions, such as plaster based on aerogels, show good potential for the thermal insulation of historic buildings [97].

### 3.3. Preservation Problems and Solutions

The globalization of cultural values and architecture has generated a standardized production far distanced from the diversity and specificity of traditional architecture. The industrialization of the building sector makes it difficult and expensive to obtain the materials or techniques traditionally used. At the same time, there is a loss of know-how and functions due to the progressive disappearance of traditional building crafts and the major abandonment of buildings caused by depopulation and changes in the lifestyle of the rural population. The lack of recognition and awareness brings somewhat of a devaluation of vernacular constructions and the consequent despoilment, vandalism, and urban speculation [98,99].

Vernacular architecture is affected by social issues, as the owners of traditional buildings are usually more interested in comfort, functionality, and maintenance than in aesthetic. Vernacular houses are not always adapted to present-day needs, and, therefore, their preservation is currently a serious challenge. High maintenance costs and, occasionally, restrictive regulations to protect heritage buildings are serious disadvantages for the inhabitants of traditional houses, driving to abandon them in favor of more modern living conditions [13,100]. Tourism offers an opportunity for the maintenance of vernacular architecture and is becoming a catalyst for economic activity and rural development [101]. However, it sometimes also becomes an obstacle to the natural modernization of traditional houses and causes constraints for owners of listed buildings [102]. Along with these risks, vernacular architecture also faces a lack of studies, measures, legislation methodologies for restoration, and dissemination, among some others.

European countries, such as Italy, the Netherlands, Spain, France, Romania, Germany, and Croatia, have made substantial efforts to preserve areas with traditional architecture by legislating and developing good practice guidelines for the restoration of vernacular buildings and organizing awareness-raising campaigns on the importance of cultural heritage aimed at the local population as the main actor to ensure its survival and preservation [103–105].

Studies focused on the resilience of vernacular constructions at risk of seismic activity, flooding, and extreme weather events have proliferated for the last two decades [106–108]. A significant number of these studies explore seismic strengthening solutions involving modern building techniques and materials. Ortega et al. [109] evaluated the efficiency of a great number of traditional strengthening solutions emerging from vernacular architecture. The assessment of architectural vulnerability to natural and anthropic risks is typically carried out by using vulnerability indexes based on the parametric and statistical analyses of the building's components [110,111]. Numerical modeling and the testing of full-scale prototypes are also powerful tools for the analysis of the seismic behavior of certain structural elements and/or construction solutions [112–115].

### 3.4. Environmental Indoor Conditions and Climate Adaptation of Vernacular Architecture

Nowadays, 20–40% of the total energy consumption in the world takes place in buildings [116]. In this context, the search for passive strategies and climate-adapted construction methods, which lead to less energy demand, fewer $CO_2$ emissions, and more comfortable indoor thermal conditions, is a research priority. Vernacular architecture is closely linked to energy efficiency due to its adaptation to climate and location* [117,118]. Climate-responsive strategies in vernacular buildings based on the use of solar radiation and wind, vegetation, or topographic conditions can buffer the indoor hygro-thermal and day-lighting fluctuations to provide thermal and visual comfort for the occupants without the need for active lighting or indoor temperature control strategies [119–121].

Over the last two decades, a large number of international studies have attempted to understand the indoor environmental conditions in vernacular architecture. By using quantitative research methods, different researchers have shown that vernacular architecture is able to provide more comfortable indoor environmental conditions and has a lower energy consumption than modern buildings [17,122,123]. Conversely, a few studies suggest that the indoor microclimate of vernacular constructions is not always thermally comfortable, especially in extreme climates with a very hot summer or a cold winter [119,124]. Some other research has stated that high relative humidity may cause several conservation problems, such as the growth of mold or visible efflorescence salts [125]. It is suggested that the use of hygroscopic building materials is a passive way to moderate indoor humidity levels.

Chandel et al. [126] identified several features that efficiently affected the indoor thermal comfort conditions in vernacular architecture that could also be adopted in modern architecture, i.e., the high thermal inertia of the wall and roof materials, orientation, floor plan, openings, and sunspaces. Solar passive features of vernacular architecture can also be used in the design of modern buildings to enhance indoor thermal comfort conditions.

Manzano–Agugliaro et al. [127] discussed the bio-climatic strategies used by traditional architecture worldwide. Although the bioclimatic solutions adopted differ from one region to another, the authors suggest that certain strategies could be exported to a different area with a similar climate. The use of these climate-adapted design strategies by the construction industry in modern buildings would promote a bioclimatic design in urban planning. Convertino et al. [128] focused on the use of a dark or clear color of the external coating as a bioclimatic vernacular strategy used in the Mediterranean region.

Studies use both quantitative and qualitative research methods, combining case studies, experimental measurements, surveys, and energy simulations to assess the thermal comfort preferences of building occupants and to produce thermal equations.

Mitigating the impacts of climate change on vernacular architecture, such as high temperatures, extreme rainfalls, or landslides, is also an emerging research topic. The impacts of climate change on the World Heritage Sites have been intensively studied by different researchers [129–131]; conversely, much less attention has been paid to vernacular architecture. Deterioration, maladaptation, and the loss of lands and loss of traditional local knowledge are some of the direct consequences that rural areas must face as a result of climate change. Another consequence of these events can be the migration of communities from rural to urban areas due to crop failures and/or land and forest degradation [132]. Increasing consequences of floods and landslides in high rainfall areas in the Eastern Black Sea Region of Turkey have been reported by Aktürk et al. [19,133]. By using aerial images and interviews with local people, they identified floods and landslide-prone areas affecting vernacular heritage sites. Anthropic influences by inadequate hydraulic works, deforestation, or bad construction practices have aggravated the effects of climate change on vernacular buildings. The authors have suggested that heritage professionals should move away from reactive to proactive planning through a combination of top-down and bottom-up approaches. Although climate change is a global phenomenon, it especially threatens vulnerable populations in third-world countries. The effects on vernacular constructions in extremely arid or cold developing regions are presented in the literature [18,34,134,135].

Although vernacular architecture is usually considered to be resilient to changes, preventive and reactive measures, such as land-use management, an appropriate site selection, water resources management, the comprehension of meteorological and biological systems, the use of locally available materials, the use of climate-adapted and durable materials, and the use of specific architectural strategies (i.e., natural ventilation, passive heating, or cooling systems) are important issues to reduce climate-related risks. The participation of the community in decision-making regarding the site, building design, and construction details is decisive for consolidating their knowledge and sense of belonging and fellowship [136].

*3.5. Reuse of Abandoned Vernacular Buildings*

The reuse of vacant vernacular buildings to accommodate new functions is a sustainable method of conservation in the medium and in the long term, provided that it guarantees the proper maintenance of the buildings [137–139]. The spontaneous reuse of abandoned buildings has occurred throughout history. In the past, the fundamental motivation guiding such reuse schemes was primarily practical and economical. However, during the 19$^{th}$ century, the concept of "heritage" was introduced, and old buildings came to be considered a reservoir of material and immaterial values. As a result, the reuse of old vernacular buildings must simultaneously combine economic interests with certain aesthetic requirements and the preservation of the significant architectural features of the buildings. As a result, the adaptive reuse of traditional buildings has become an interdisciplinary academic field in its own right, involving experts from different fields, such as architects, archaeologists, urban planners, engineers, and interior designers [140].

When deciding how to reuse a vernacular building, representatives of the administration, promoters, and owners often have different ideas. Municipal architects must ensure that regulations are met and that authenticity is maintained, while contractors and developers may be more concerned with reducing costs and achieving an adequate level of comfort. Therefore, in order to make an appropriate decision between various reuse proposals, multiple factors have to be taken into account. The suitability of redundant buildings to adopt new uses without losing their original character can be assessed by multicriteria decision-making (MCDM) techniques based on the assessment of different viable options with respect to multiple properly weighted criteria [141–143]. The Analytic Hierarchy Process (AHR), a method for organizing and analyzing complex decisions based on pairwise comparisons and linear algebra, was also successfully used to identify the best hypotheses for the adaptive reuse of vernacular buildings [144].

According to García and Ayuga [145], the conversion of abandoned vernacular buildings in rural areas results in significant benefits, such as energy- and materials- saving, job creation, the promotion of cultural tourism, the preservation of valuable ethno-cultural resources, the recovery of traditional construction techniques, the creation of community consciousness, and a more pleasant image of the villages and rural landscapes. Nevertheless, Verhoeve et al. [146] argue that in peri-urban or tourist interest areas, the hidden re-use of rural buildings by nonagricultural activities may result in a loss of the agricultural character of these areas and should be legally quantified and controlled. The development of design guidelines to carry out conversion schemes is of undoubted interest, although too restrictive of regulations may result in the loss of interest by owners and the continued decline of this heritage.

## 4. Conclusions

Vernacular architecture constitutes a heritage resource of undoubted interest that is worth promoting and preserving. Despite its tangible and intangible value, its survival is deeply threatened by factors such as the industrialization and globalization of modern construction techniques, the loss of the use of traditional buildings, which sometimes do not meet the original functions or the level of comfort currently required, the urban development pressure in the more densely populated areas, and the occasional existence of excessively restrictive regulations.

In the upcoming decades, the consequences of climate change (i.e., intense rainfall, droughts, floods, severe snowfalls, dust storms, etc.) are expected to intensify in rural areas, with negative effects on crops and vernacular constructions. This represents an emerging risk that urgently needs to be quantified and investigated. The production of new construction materials compatible with old buildings or the modernization of some traditional solutions, such as fiber-reinforced earth, the publication of design guides for the rehabilitation of traditional buildings, and the launching of social awareness campaigns, are the main instruments for their revaluation.

In recent years, researchers from all over the world have produced literature aimed at assessing the architectural features of traditional buildings, analyzing the spatial patterns, surveying the climatic conditions inside the buildings, or studying the thermal passive or bioclimatic principles in the vernacular constructions applicable to modern buildings.

According to the literature, geographic information systems and satellite images are valuable tools for locating vernacular buildings, studying their spatial distribution, analyzing typologies, and managing conservation priorities. Consequently, their use is expected to expand in the coming years. Computer numerical modeling is also being successfully used to simulate the structural behavior of building components and to study the hygro-thermal behavior inside the buildings or in the seismic vulnerability assessment, among other applications. To advance the goal of preserving vernacular architecture in the coming centuries, research should continue in interdisciplinary teams, promoting studies in less studied areas (i.e., Africa, South America, or Australia), providing new tools and theoretical bases for the conservation of traditional buildings, and incorporating modern solutions in old vernacular buildings to satisfy current comfort standards without excessively changing their original character.

Examples of vernacular architecture across the world highlight the possibility of using natural resources in an efficient way and adopting climate-adapted strategies to achieve human comfort. The architectural, cultural, and ethnological values of vernacular buildings confirm the need to preserve them for future generations. It is hoped that the results of this study will serve to identify some of the major challenges to be faced by vernacular architecture in the current global context (i.e., industrialized production techniques, the loss of traditional knowledge, the loss of identity, etc.) and to highlight some research topics concerning vernacular architecture for the coming decades (inventory and cataloging techniques, the development of new building materials, the search for passive energy-saving strategies, or the adaptation to the effects of climate change, among others).

Nevertheless, the present and future sustainability of vernacular architecture will only be possible through a solid knowledge of the relationship of the buildings with local cultures, lifestyles, climates, environments, and economies. This knowledge will provide an essential basis for integrating the principles of vernacular architecture into modern design, leading to more resilient buildings adapted to the needs of dwellers and local conditions. The methods and tools used are only important to the extent that the true identity of vernacular architecture can be captured.

**Funding:** This research received no external funding.

**Data Availability Statement:** Not applicable.

**Conflicts of Interest:** The author declares no conflict of interest.

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
