# Peer review of "Challenges and Current Research Trends for Vernacular Architecture in a Global World: A Literature Review"

_buildings, doi:10.3390/buildings13010162_

Round 1

Reviewer 1 Report

The hardest part about writing on vernacular architecture is its specificity and local/regional variation across the globe. This makes any broad paper like this problematic in that it rarely gets into any depth, which is the attractor of the individual components of a topic. This reads like a broad introduction to a larger book, where the points are of interest, but never developed in a piece of this length. The conclusions actually raise better problems than the introduction suggests are coming, but the brevity of the paper length limits their impact. I would narrow the scope to those conflict points as suggesting need for research versus an overview, or how some research is focused on conflicts or problems.

In terms of structure, I think section 2.2 is a better lead topic than 2.1. Section 2.1 seems particularly confused with regard to conflicting issues of representation, typology, and publication topics. Again, each of these has more than enough content for a whole paper.

I think the paper does raise important points in challenges to the vernacular that would be basis of their own papers.

I would contest the initial point that all vernacular building falls in "model of sustainability". Certainly some do, based on local custom as to what is ethical and sustainable. This is primarily in the area of thermal comfort, heating sources not being sustainable [burning peat, coal, oil, or wood], but also changing nature of dwelling in terms of family structure and gender roles. Eventually in lines 178-188 this dichotomy is finally addressed but could be seeded up front as a challenge to traditional models of vernacular dwelling.

The emphasis on the paper is quantitative, but an acknowledgment of the distinct qualitative aspects that may be more important [perhaps outside the limits of this paper] should be made, especially as that is a key to the economic preservation of many examples due to heritage tourism.  It begins to be addressed in a cursory manner in section 2.3.

The point on a main attribute of "active participation of users" [lines 45-46]:  increasingly outstanding examples are taken away from local actors and a kind of natural evolution and given into control by experts and inclusion in special districts limiting transitions or adaptations. There is little The conclusion badly touches on sustainability or emerging impacts of global warming, especially in places where vernacular works are threatened, which is another dire need for cataloguing and research.

The point on traditional materials being less labor intensive that modern ones is rather counter-intuitive and not able to be explained by the author, while the carbon footprint is adequately supported. 

One of the most interesting points [that could be a whole paper] is "However, the use of an environmentally friendly material in a certain local area may be a threat in another one. ". That is a novel and emerging point worthy of expansion.

The references are quite impressive, but here are some additional ones below that may be of help. The Rapoport work is of the same era as Rudolfsky, but went more into depth of why and how things developed the way they did, especially for traditional dwelling. 

There is a lack of dressing ongoing Indigenous dwelling, particularly in Africa and South America in the work.

Some helpful comparative references:

Elizabeth Golden, Building from Tradition: Local Materials and Methods in Contemporary Architecture, Routledge, 2017.

Ahmad Hamid, Hassan Fathy and Continuity in Islamic Arts and Architecture: The Birth of a New Modern, American Univ. in Cairo, 2014.

Allen Noble, Traditional Buildings: a Global Survey of Structural Forms and Cultural Functions, I.B. Tauris, 2007.

Amos Rapoport, House Form and Culture. Prentice– Hall, 1969.

 Ali Sayigh [ed.], Sustainable Vernacular Architecture, Springer, 2019.

Author Response

I would like to thank reviewers for their valuable suggestions that certainly improve the content of the manuscript. A point-by-point response to the reviewer’s comments is included as a PDF file.

Reviewer 2 Report

I would like to thank the author for his contribution. I have comments for author to improve his paper.

The title of the paper should be revised because there is another paper under the title Research Trends in Vernacular Architecture: A bibliometric study

Abstract does not mention the methodology of this paper. Lines 13-16 can reveal that this study is qualitatively analyze the existing literature on…

Table 1 should be revised. This table is not really informative. There is no mentioning of inclusion and exclusion criteria. The author can from line 86 put the title of methods. In this section, he can explain that he has taken systematic literature as a methodology by giving references to the sources. Then in a subsection of document selection he can describe inclusion and exclusion criteria. Please check Review paper How Is Australia Adapting to Climate Change Based on a Systematic Review? On sustainability journal. Author can adopt this exclusion-inclusion table here. Then he needs to explain why he included books, reviews and articles. Reviews and books are usually not included I  this type of methodology. Is there a reason for their inclusion? Also are these sources in English or Spanish sources are included? Please justify the selection. Then data analysis needs to be explained. Is it analyzed quantitatively with numbers and percentages or qualitatively? Are they analyzed thematically? Another thing is why the author did not include vernacular heritage in his search terms.

Page 2 in the introduction, the author should also mention climate resilience and adaptation of vernacular architecture as an opportunity for future.  

Page 3 line 88 author should state when he conducted this literature review in month and year. Because these results vary depending on the date. The fact that 2021 is not taken may be related to that.

Figure 1 includes 2 figures and below it says Peoples China and China. This mistake should be corrected.

2.2. Typological analysis of vernacular buildings does not really include typology in it.

2.4. Climate adaptation misses important literature there because the work from Aktürk and Fluck fall under this theme. Similarly, she has many other works that fall under this theme but doesn’t appear in literature analysis? Detection of disaster-prone vernacular heritage sites at district scale: The case of Fındıklı in Rize, Turkey should be surely in this analysis.

Titles are very generic. Just climate adaptation doesn’t say much but climate adaptation of vernacular buildings is more clear. Similarly, 2.5. Reuse of abandoned traditional buildings would be better but then why using traditional buildings instead of vernacular. The author should be consistent with the use of term throughout the text.

Despite the great number of included documents in the literature analysis there is only 118 references. The author should give references to all the sources he analyzes.

Author Response

(The authors gave the same response as above.)

Reviewer 3 Report

The paper discussed the topic of research trends in vernacular architecture, which is an interesting and important topic to determine previous and current research trends in this topic and various research methods according to different topics and geographical regions; however, some comments must be addressed to improve the paper's quality for journal standards, such as:

- This article was written in a brief manner, which does not suit review articles, and in order to be more comprehensive and beneficial in presenting and analyzing all previous research works, it is recommended to do a filter for literature research using one of the visual filtering tools, with a focus on the filter of these literature according to the most important main topics.

- It is preferable to clarify which purposes have been discussed in relation to the issue of vernacular architecture according to the various specialties.

- I believe that a discussion section should come before the conclusion section in order to provide a more in-depth vision of the different directions and the most important research purposes that covered them, as well as compare the highlight results of the papers according to the research methods used, as well as determine the shortcomings and recommendations presented for future studies.

- The author noted in the conclusion and recommendations of the importance of strengthening studies in less studied regions, such as (To advance the goal of preserving vernacular architecture in the coming centuries, research should continue in interdisciplinary teams promoting studies in less studied areas (i.e. Africa, South America or Australia), and he did not explain this result within this paper, so it is preferable to add it to filtering the studies according to the geographical scope, to provide a new additional value on the amount of research according to the geographical scope and possibly the subjects.

- The recommendations for future research (Several tools such as geographic information systems, satellite images, or computer numerical modeling can be very useful for the study and management of vernacular buildings) appear logical and good; the priorities on which these recommendations were based on the content of this paper were not clarified.

Author Response

(The authors gave the same response as above.)

Round 2

Reviewer 1 Report

The material reads better. It is always difficult to judge the efficacy of an introductory chapter without seeing the specifics of the more developed focus chapters. I think this positions the materials better and is valuable.

Author Response

Thank you again for your helpful and constructive review.

Reviewer 2 Report

I would like to thank author for his revisions and improvements. I have just minor remarks for the paper. Why use of asterisk in the second group of words in parenthesis and not in vernacular and traditional. This is not consistent in terms of search terms.

“Vernac*” OR “tradit*”) AND (“archit*” OR “build*” OR “herit* 

Line 165 in the sentence ‘However, it should be noted that there are large areas of the world whose vernacular architecture has been little studied or at least is little known at the international level’ should be revised as However, it should be noted that the studies on vernacular architecture have been insufficient in many countries at the international level’

Line 339 references in brackets 19-128 mean that sources between these numbers. It should be corrected as 19,128.

Lines from 348 to 355 every sentences are written in italics. There must be a mistake here. Please make sure that this is corrected.

Lines between 399 and 402 seems to be a paragraph taken without giving a reference Vernacular Heritage as a Response to Climate or Detection of disaster-prone vernacular heritage sites at district. Similarly 412-414 this line is again the use of geographic systems in vernacular heritage is a reference to Detection of disaster-prone vernacular heritage sites at district.

I still think that conclusion is weak. Because it says the studies are mostly based on quantitative. And based on these countries… yes but then what needs to be done or what are the suggestions of the author. Does he suggest that the studies should consider qualitative approaches together with quantitative? The author mentions climate threats on vernacular but then what methods or approaches future researchers should focus on? 

The author states 'to reduce the number of documents to a reasonable number.' in line 144 but this is not a reasonable explanation as the systematic literature review is naturally being done in great numbers and if necessary with the help of second or third authors. It can be done 350 documents or more. This part of the sentence should be removed.

Last comment is that there are much more extensive reviews being done in literature on the topic. For instance, if meta-analysis is being done or not. If yes why and if no why not. There is still no explanation on why these themes are selected. Was it content analysis of the selected literature? And the limitations of this method is not explained. Non English sources could shed light into some of the studies on vernacular architecture but due to the misinterpretations that could arise from the translation that these sources are not translated. something like this can be explained. There is also not a document type criteria inclusion or exclusion does this mean that only articles are selected? If so why or why not. These are not explained. If the analysis method is not explained the methodology of this paper will not be rigorous enough.

Instead of giving major revisions again, I would suggest author to take his time to really revise it well and take a look at other review examples on how other authors gave precision to its methodology. If the author would like to be cited and referred by other scholars, it is highly recommended to make it as clear as possible. I also saw English mistakes but I trust that author will check these really carefully before its publication.

Author Response

I would like to thank the reviewer again for his/her valuable suggestions. A point-by-point response to the reviewer’s comments is attached as a PDF file.

Round 3

Reviewer 2 Report

I trust the adjustments from the author so I accept the publication of the paper.

Author Response

Thanks for your valuable suggestions.